# The Effects of Catabolism Relationships of Leucine and Isoleucine with *BAT2* Gene of *Saccharomyces cerevisiae* on High Alcohols and Esters

**DOI:** 10.3390/genes13071178

**Published:** 2022-06-30

**Authors:** Lin Zhang, Yiqian Zhang, Zhongqiu Hu

**Affiliations:** 1College of Food Science and Engineering, Northwest A&F University, Yangling, Xianyang 712100, China; xslj@nwafu.edu.cn (L.Z.); zhangyiqian@nwafu.edu.cn (Y.Z.); 2Laboratory of Quality & Safety Risk Assessment for Agro-Products, Ministry of Agriculture, Yangling, Xianyang 712100, China; 3National Engineering Research Center of Agriculture Integration Test Yangling, Xianyang 712100, China

**Keywords:** leucine, isoleucine, *BAT2* gene, higher alcohols and higher esters, relationship, esterification reaction balance

## Abstract

This study sought to provide a theoretical basis for effectively controlling the content of higher alcohols and esters in fermented foods. In this work, isoleucine (Ile) or leucine (Leu) at high levels was used as the sole nitrogen source for a *BAT2* mutant and its parental *Saccharomyces. cerevisiae* 38 to investigate the effects of the addition of amounts of Ile or Leu and *BAT2* on the aroma components in the flavor profile using gas chromatography mass spectrometer (GC-MS). The results showed that 2-methyl-butyraldehyde, 2-methyl-1-butanol, and 2-methylbutyl-acetate were the products positively correlated with the Ile addition amount. In addition, 3-methyl-butyraldehyde, 3-methyl-1-butanol, and 3-methylbutyl-acetate were the products positively correlated with Leu addition amount. *BAT2* deletion resulted in a significant decline in the yields of 2-methyl-butyraldehyde, 3-methyl-butyraldehyde,2-methyl-1-butanol, and 3-methyl-1-butanol, but also an increase in the yields of 2-methylbutyl-acetate and 3-methylbutyl-acetate. We speculated that *BAT2* regulated the front and end of this metabolite chain in a feedback manner. Improved metabolic chain analyses, including the simulated energy metabolism of Ile or Leu, indicated that reducing the added amount of branched-chain amino acids, *BAT* mutation, and eliminating the role of energy cofactors such as NADH/NAD+ were three important ways to control the content of high alcohols and esters in fermented foods.

## 1. Introduction

Branched-chain higher alcohols such as isoamyl alcohol, isobutanol, and isoamyl acetate are the key flavor components and esters in many fermented foods [1,2]. An appropriate content of higher alcohols and esters makes fermented foods mellow, soft, full, and harmonious; however, too high a content often gives such fermented foods a fusel oil taste, delivering a strong punch feeling to consumers [3]. Therefore, controlling the content of these is important for promoting the quality and popularity of such foods. Previous studies showed that higher alcohols were derived from the degradation of branched-chain amino acids (BCAAs) in *Saccharomyces cerevisiae* via the so-called Ehrlich pathway [4,5]. Both *BAT1* and *BAT2* genes were inferred to be involved in the metabolism of BCAAs, as the encoded Bat1p and Bat2p transaminases corresponding to the two genes catalyzed the deamination of BCAAs, which was thought to be the first step in the degradation of BCAAs [2,5,6]. It was reported that Bat1p was located in the mitochondrion and Bat2p existed in the chromosomes of a yeast cell. The deletion of any particular *BAT* gene did not impair the growth of yeast cells, but the deletions of both of the two genes resulted in an auxotrophy strain of BCAAs, while a severe growth reduction was observed on glucose-containing media [7]. Therefore, the double-gene-deletion mutant strain is of less interest in controlling the negative effects of the addition of BCAAs on the higher alcohols due to the growth retardation [3]. The studies of a single-gene-deletion mutant strain would be meaningful to reduce the production of higher alcohols to a suitable content in order to promote the quality and popularity of fermented foods. The addition of specific amino acids has targeted aromatic effects on final alcoholic fermentation products because these amino acids are the precursors of some aromatic compounds. Adding amino acids into media is a popular technique in the modern fermentation industry to improve the aroma quality of fermented foods [8]. Isoleucine (Ile) and leucine (Leu) are typical representatives of BCAAs among assimilable nitrogen sources in food matrices; they are isomer precursors of higher branched-chain alcohols. It was reported that the formation of yeast biomass and the fermentation rate were regulated by the amount of BCAAs, which thereby ultimately affected the composition of aroma compounds [8]. Although the influence of BCAAs mostly at low levels on production of branched-chain higher alcohols has been reported; the researchers concluded that different assimilable nitrogen source levels might have had different effects on the pathway [9], but the mechanism of feedback and/or repression of the metabolite chain is still not well understood, and knowledge of the quantitative influence of BCAA additions and *BAT2* deletion on the yields of branched-chain higher alcohols and esters is especially lacking. In this work, the effects of Ile or Leu addition at high levels and *BAT2* deletion on high alcohols and esters derived from Ile or Leu were investigated by respectively using Ile or Leu as the sole nitrogen source of both a *BAT2* mutant *S. cerevisiae* 38 and its parental *S. cerevisiae* 38, in which the material flow and energy flow were reasoned. In addition, the feedback mechanisms of *BAT2* gene were deduced to gain a new understanding of the catabolism relationships of Leu and Ile with the *BAT2* gene of *S. cerevisiae* to high alcohols and esters in the flavor components, as well as to provide a theoretical basis for effectively controlling the content of higher alcohols and higher esters in fermented foods.

## 2. Materials and Methods

### 2.1. Materials

#### 2.1.1. Strain and Vector

A cider yeast *S. cerevisiae* 38 used was from the authors’ collection [10]. The vector pUG6 was purchased from BioVector National Type Culture Collection Inc. (Beijing, China), and contained a *loxP-kanMX-loxP* module for gene deletion. The used primers, which are listed in Table 1, were all synthesized by Beijing Sunbiotech Co., Ltd. (Beijing, China).

#### 2.1.2. Enzyme and Reagents

The rTaq DNA polymerase, Takara MiniBEST Agarose Gel DNA Extraction Kit Ver. 3.0, ss-DNA, and RNase A were provided by Takara Bio Inc. (Tokyo, Japan). A yeast DNA Kit used was the product of Omega Bio-tek Inc. (Feyou Biotechnology, Shanghai, China). Lithium acetate dehydrate was purchased from Guangzhou Jinhauda Chemical Co., Ltd. (Guangzhou, China). Polyethylene glycol (MW4000), cetyl-tri-methyl-ammonium bromide (CTAB), Tris-HCl, sodium acetate, ethylene-di-amine-tetra-acetic acid (EDTA), ethidium bromide (EB), and agar were obtained from Sanli Chemical Company (Yangling, China). Bacto-yeast extract, bacto-peptone, and dextrose were purchased from Beijing Aoboxing Bio-tech Co. Ltd. Analytically pure L-Leu, L-Ile, standard hexaldehyde, standard isoamyl alcohol, standard 3-methylbutyl acetate, G4l8, and ampicillin were supplied by Sigma-Aldrich Corporation (Beijing, China).

### 2.2. Cloning and Sequencing of the BAT2 Gene of S. cerevisiae 38

Strain 38 was cultivated overnight at 28°C and 120 rpm in yeast peptone dextrose (YPD) medium, which contained 1% yeast extract, 2% Bacto peptone, and 2% dextrose. Cells were collected by centrifugation at 5000 rpm for 5 min at room temperature. The genomic DNA was isolated with a Takara MiniBEST Agarose Gel DNA Extraction Kit according to the instructions, and was used as the PCR amplification template for the *BAT2* gene. The used primer pair was Fw-1 (34 bp) and Rv-2 (32 bp), as shown in Table 1, which were homologous to the up-flanking (280~258 bp) or down-flanking (249~269 bp) fragment of the *BAT2* gene, respectively. The PCR reactant measured 50 μL (containing 1 μL 10 μM Fw-1, 1 μL 10 μM Rv-1, 2.5 μL DMSO, 5.0 μL 10×PCR buffer, 1.6 μL 25 mM MgCl_2_, 1 μL 2.5 mM dNTP mixture, 5.0 μL DNA template, 0.3 μL rTaq enzyme, and nuclease-free water added up to 50 μL) and was carried out in the Bio-Rad^®^ ALD1244 DNA Engine (Bio-Rad, Hercules, CA, USA). The optimum PCR protocol was as follows: the initial denaturation temperature was 94 °C for 2 min; then 40 cycles in which each cycle consisted of 94 °C for 20 s, 55 °C for 20 s, and 72 °C for 1 min; then a final extension at 72 °C for 10 min. The PCR product was sequenced by Sangon Biotech (Shanghai, China).

### 2.3. Construction of the BAT2 Gene-Deletion Cassette through LPSHR

The construction method of the *BAT2* deletion cassette was a long primers–short flanking homology region (LPSHR) [11,12], which required the design of a long primer pair. The designed long forward primer Bat2-deletion-FW and long reverse primer Bat2-deletion-RV are both shown in Table 1.

As shown in Figure 1, the forward primer *BAT2* deletion-FW (64 bp) contained Segments C (45 bp) and E (19 bp); Segment C was homologous to the up-flanking of the *BAT2* gene of *S. cerevisiae* 38, and Segment E was homologous to the front sequence of the loxp-KanMX-loxp module of pUG6. The reverse primer Bat2-deletion-RV (67 bp) contained Segments D (45 bp) and F (22 bp); Segment D was homologous to the down-flanking of the *BAT2* gene of *S. cerevisiae* 38, and Segment F was homologous to the end sequence of the loxp-KanMX-loxp module of pUG6. It was seen that the deletion cassette (C+E+ loxp-KanMX-loxp+F+D) of *BAT2* could be amplified by PCR using the long primer pair shown in Figure 1. During the PCR process, the vector pUG6 was used as the template, and the recipe for the 50 μL of PCR reactant was as mentioned above, except for different templates and primers. The optimized PCR program was as follows: 95 °C for 2 min in the initial denaturation step; followed by 40 cycles in which each cycle consisted of three stages, i.e., at 94 °C for 20 s, 72 °C for 1 min with a 0.2 °C decline per cycle, and 2 min at 72 °C; with a final extension at 72 °C for 10 min after the 40 cycles had ended. The obtained PCR products were checked using 1% agarose gel electrophoresis to verify the construction validity of the *BAT2* deletion cassette. The verified *BAT2* deletion cassette was transformed into *S. cerevisiae* 38 to delete the *BAT2* gene and to obtain the *BAT2* mutant *S. cerevisiae* 38 (Δ*BAT2*).

### 2.4. Deletion of the BAT2 Gene of Parental S. cerevisiae 38

#### 2.4.1. Transformation of the BAT2 Deletion Cassette

The transformation of the *BAT2* deletion cassette was carried out using the lithium acetate method as described in previous papers [13,14] with certain modifications. In detail, the cultured cells of parental *S. cerevisiae* 38 were collected through centrifugation at 4500× *g* at 4 °C for 10 min. The precipitated cells were resuspended in 1.5 mL of sterile 1.0 M TE/LiAc and centrifuged at 5000 rpm at 4 °C for 1 min again, then the cell pellets were resuspended with 100 μL of sterile 1.0 M TE/LiAc (over 2 × 10^9^ cells/mL), which then were mixed in turn with 300 μL of chilled 40% PEG4000, 36 μL chilled 1.0 M LiAc, and 5 μL of 10 mg/mL ss-DNA preboiled for 20 min, and finally 34 μL *BAT2* deletion cassette DNA (~550 ng/μL) was added and mixed well. The obtained transformation mixture was incubated in a 30 °C water bath for 30 min and promptly heat-shocked at 42 °C for 20 min, then was uniformly mixed with 800 μL of YPD medium and centrifuged at 12,000 rpm for 10 s at 4 °C. Subsequently, the collected cell pellets were resuspended with 1 mL of YPD and incubated for 2 h at 30 °C. After that, the incubated mixture was centrifuged at 4 °C at 12,000 rpm for 10 s again, and then the cell pellets were immediately resuspended with 0.2 mL of YPD medium and evenly plated onto a YPD plate containing 200 μg/mL of G418 and cultured at 28 °C to select potential transformants. Three independent transformation cultures were carried out for biological repeats.

#### 2.4.2. Selection and Verification of the Transformants

From the third to fifth day, colonies obviously grew in the plates with 200 μg/mL G418; these were the potential transformants. They were picked out with an inoculation ring and transferred to a new plate containing 200 μg/mL of G418 to be secondly selected, while the untransformed parental colonies were used as the negative controls. The colonies that could continue to grow in these new plates became G418-resistant transformants in the next three days, and were the potential transformants. Then, the potential transformants were separately picked and cultured in liquid YPD media with 50 μg/mL of G418 at 28 °C with shaking at 120 rpm until OD_600nm_ of the culture was equal to 600. The genomic DNA of every potential transformant was extracted and purified in turn using an Omega Yeast DNA Kit according to the instruction manual provided by the supplier. The obtained genome DNA of every potential transformant was used as a PCR template to determine whether their *BAT2* genes were correctly deleted. The primers Fw-bat2-2 and Rv-bat2-2 listed in Table 1 were utilized as the verified primer pair. The PCR program consisted of three steps: the first step was an initial denaturation at 94 °C for 2 min; the second step included 40 cycles in which each cycle consisted of 20 s at 94 °C, 20 s at 45 °C, and 1 min at 72 °C; and the third step was a final extension at 72 °C for 10 min. The PCR products were checked using 1% agarose electrophoresis analyses to identify whether the *BAT2* genes of the potential transformants were correctly deleted (*n* = 3). 

### 2.5. Metabolites Analyses of Ile and Leu as Sole Nitrogen Sources

#### 2.5.1. Design of the High-Concentration Gradient Supply of Ile and Leu in Media

Synthetic complete dextrose medium without amino acids (SCD: 0.7% yeast nitrogen base without any amino acids, 2% dextrose, pH 5.5) was used as a control medium. M1 (SCD added to 0.5 M Ile without other amino acids) and M2 (SCD added to 1.0 M Ile without other amino acids) were respectively used as the gradient addition media for Ile. M3 (SCD added to 0.5 M Leu without other amino acids) and M4 (SCD added to 1.0 M Leu without other amino acids) were respectively used as the gradient addition media for Leu. The fermentation experiments on the *BAT2* mutant *S. cerevisiae* 38 (Δ*BAT2*) and its parental *S. cerevisiae* 38 were carried out in these media at 28 °C with shaking at 500 rpm. All trials were conducted in triplicate.

#### 2.5.2. Metabolite Analyses

The analyses of metabolites were performed using gas chromatography–mass spectrometer (GC-MS). In advance, a headspace solid-phase microextraction (SPME) procedure was carried out according to the studies in [15,16] with a slight modification prior to the GC-MS analysis. In detail, 7 mL of fermentation liquor from every trial was firstly mixed well with 1.0 g of anhydrous sodium sulfate in a consecutively numbered 15 mL glass vial sealed with a silicone-septum cap (Supelco), automatically stirred with a small polytetrafluoroethylene (PTFE)-coated bar and maintained at 55 °C on a Reacti-Therm heater (Pierce, Rockford, IL, USA). A polydimethylsiloxane fiber needle (10 mm long, 100 μm in diameter; Supelco of USA) was pierced into the vial through the cap after 10 min when the vapor–liquid equilibrium was obtained, and exposed in the vapor phase of the vial to adsorb metabolites for 40 min. Then, the fiber needle was pulled out from the vial and immediately inserted into the GC injection port at 230 °C and exposed there for 3 min to completely desorb the metabolites. The GC-MS analysis was performed with a ISQ single quadrupole GC-MS (Thermo Fisher scientific, Waltham, MA, USA), equipped with a DB-WAX capillary column (60 m × 0.25 mm, ID 0.25 μm film thickness; Agilent Technologies, Santa Clara, CA, USA) and a mass spectrometer (MS) detector; the used analysis method was modified according to previous reports [17,18,19,20,21]. In detail, the initial oven temperature was held at 40 °C for 2.5 min, then increased at a rate of 5 °C/min until reaching 200 °C, and then held for 6 min. Subsequently, the temperature was raised to 250 °C at a rate of 10 °C/min, then maintained at 250 °C for another 5 min. Helium was used as the carrier gas, and the flowrate was 1.0 mL/min. Analytes were injected in the split mode. The mass spectrometer was set at 70 eV in electron-impact (EI) mode. The mass-to-charge ratio (*m/z*) ranged from 33 to 450 atomic mass units (AMUs). The temperature of the ion source and quadrupole was set at 200 °C and 150 °C, respectively. Peak identifications of the volatile components were achieved via comparison of the mass spectra with the mass spectral data from the NIST 2011 libraries. Standard hexanal was used as the external reference standard to quantify aldehyde metabolites (standard curve correlation coefficient *R*^2^ = 0.992). Standard isoamyl alcohol was used as the external reference standard to quantify higher alcohol metabolites (standard curve correlation coefficient *R*^2^ = 0.999). Standard 3-methylbutyl acetate was used as the external reference standard to quantify ester metabolites (standard curve correlation coefficient *R*^2^ = 0.995). The analyses of Ile and Leu consumption in media were carried out using RP-HLPC according to authors’ previous paper [22].

#### 2.5.3. Energy Metabolism Analysis

The steric energies of the relevant metabolites were calculated using an MM2 molecular force field according to previous papers [23,24] with a commercial software package (CS ChemBats3D Pro 4.0 supplied by the Cambridge Soft Corporation). Each energy value was obtained through at least 10,000 iterations and optimizations.

## 3. Results

### 3.1. Sequencing and Deletion of the BAT2 Gene Using the LPSHR Method

The *BAT2* gene of the parental *S. cerevisiae* 38 was sequenced by Shanghai Sangon Biotech of China; its size was proofed to be 1131 bp, which laid the groundwork for determining the length of the *BAT2* gene-deletion cassette. According to our simulation calculation using Clone Manager 7.1 software, the PCR product size of the deletion cassette should have been 1703 bp. As Figure 2A shows, a ~1700 bp bright band of PCR product showed that the *BAT2* deletion cassette had been successfully constructed. After transformation and screening, there were 10 potential transformants growing on an incubation YPD plate screened with 200 μg/mL of G418, as shown in Figure 2B; these in turn were named 1, 2, 3, p1, p2, p3, p4, p5, p7, and p8. We observed that these 10 potential transformants could grow normally on the new screening YPD plate that contained 200 μg/mL of G418, but the negative controls could not grow (N1 and N2 in Figure 2B). After PCR verification, we found that only four transformants (p1, p4, 2, and 3) were true deletion transformants because no *BAT2* bands were amplified, but the others were false-positive transformants, as displayed in Figure 2C (*BAT2* had not been properly deleted). The true transformant p1 was named *S. cerevisiae* 38 (Δ*BAT2*) and selected to further investigate the effect of the *BAT2* gene on the catabolism of Leu and Ile.

### 3.2. Catabolic Analyses of Ile and Leu

#### 3.2.1. Catabolic Analyses of Ile

The catabolic analyses of Ile were qualitatively and quantitatively analyzed using GC-MS by comparing the metabolites produced by the *BAT2* mutant *S. cerevisiae* 38 (Δ*BAT2*) and parental *S. cerevisiae* 38 for 12 h in media M1 (0.5 M Ile+SCD without other amino acids) and M2 (1.0 M Ile+SCD without other amino acids), as well as the control group (SCD without any amino acids). The compounds among the metabolites that positively correlated with the Ile addition were only selected for reporting. As Figure 3A shows, 2-methyl-butyraldehyde was not detected in the control group of the two strains, but was detected in the M1 and M2 groups. The mass spectrogram of 2-methylbutyraldehyde is displayed in Appendix A. We observed that the 2-methyl-butyraldehyde yields of *S. cerevisiae* 38 (Δ*BAT2*) and parental *S. cerevisiae* 38 significantly increased with the initial concentration of Ile in both M1 and M2 compared to the control group (*p* < 0.01), which roughly presented a proportional relationship between the yield of 2-methyl-butyraldehyde and the addition amount of Ile. On the other hand, the 2-methyl-butyraldehyde yields of *S. cerevisiae* 38 (Δ*BAT2*) were significantly less than those of parental *S. cerevisiae* 38 in M1 and M2 (*p* ˂ 0.01, *n* = 3); these were 3.81 ± 0.57 μM in M1 and 7.3 ± 1.01 μM in M2 for *S. cerevisiae* 38 (Δ*BAT2*), and 9.87 ± 0.69 μM in M1 and 21.2 ± 1.49 μM in M2 for Parental *S. cerevisiae* 38. This indicated that the deletion of *BAT2* led to a decrease in the yields of 2-methyl-butyraldehyde, as the existence of the *BAT2* gene upregulated the conversion rate of Ile to 2-methyl-butyraldehyde. It was calculated that the deletion of the *BAT2* gene led to a 2.59~2.9-fold decrease in the yields of 2-methyl-butyraldehyde. This finding was consistent with previous reports [2,4]. The regulation of initial Ile concentration appeared to be independent of regulation of the *BAT2* gene. Because the tendency was always toward the higher initial concentration of Ile in vitro for the yeast cells, the higher yields of 2-methyl-butyraldehyde for the two strains resulted. It seemed to have nothing to do with the existence of the *BAT2* gene, which suggests that the initial concentration of Ile may have played a critical role in giving a pressure signal to the cells in vitro. This was consistent with the finding that the positive effect of BCAA addition on major volatile compounds in wine were largely associated with the increment in amino acid transportation (upregulation of *GAP*) and the yeast population [8]. These results indicated that the initial concentration of Ile and *BAT2* gene jointly upregulated the production of 2-methyl-butyraldehyde in the parental *S. cerevisiae* 38. In short, we inferred from the above results that 2-methyl-butyraldehyde was one of decomposition products of Ile, which was consistent with previous findings [6,9,25,26].

As Figure 3B shows, another product positively correlated with Ile addition as 2-methyl-1-butanol. The mass spectrogram of 2-methyl-1-butanol is shown in Appendix A. As shown in Figure 3B, the yields of 2-methyl-1-butanol significantly rose with an increase in the initial Ile concentration for the two strains in M1 and M2 compared to the control media (*p* < 0.01, *n* = 3). The tendency was always toward a higher initial concentration of Ile in the media, resulting in the higher yields of 2-methyl-1-butanol for the two strains. The regulation of initial Ile concentration appeared to be independent of regulation of the *BAT2* gene. According to the direction of the Ehrlich pathway [2,5,6], 2-methyl-1-butanol should be converted from 2-methyl-butyraldehyde, which is a reduction reaction. Although the yields of 2-methyl-1-butanol also significantly increased with the increase in 2-methyl-butyraldehyde, the yields of the former were ~20 times as much as those of the latter. This may have been due to the fact that 2-methyl-butyraldehyde is very reactive and can be easily reduced to 2-methyl-1-butanol. On the other hand, whether in the M1 (0.5 M Ile+SCD) or M2 (1.0 M Ile+SCD) medium, the yields of 2-methylbutanol for the mutant *S. cerevisiae* 38 (Δ*BAT2*) were both significantly lower than those of the parent strains (*p* ˂ 0.05, *n* = 3), which reached 0.11 ± 0.01 mM in M1 and 0.24 ± 0.019 mM in M2 for *S. cerevisiae* 38 (Δ*BAT2*), and 0.24 ± 0.04 mM in M1 and 0.40 ± 0.07 mM in M2 for parental *S. cerevisiae* 38. This indicated that the deletion of *BAT2* resulted in a decrease in the 2-methyl-1-butanol yields; and vice versa, the 2-methyl-1-butanol yields were upregulated by both *BAT2* and initial Ile concentration in the parental *S. cerevisiae* 38.

The third product positively correlated with Ile addition was 2-methylbutyl acetate. As Figure 3C shows, no 2-methylbutyl acetate products were detected in any control media for the two strains. However, in M1 (0.5 M Ile+SCD) and M2 (1.0 M Ile+SCD), the yields of 2-methylbutyl acetate of the two strains were positively correlated with the initial Ile concentrations, which reached 0.072 ± 0.011 μM in M1 and 0.11 ± 0.017 μM in M2 for the mutant *S. cerevisiae* 38 (Δ*BAT2*), and 0.06 ± 0.003 μM in M1 and 0.073 ± 0.005 μM in M2 for the parental *S. cerevisiae* 38. We observed that the 2-methylbutyl acetate yields of *S. cerevisiae* 38 (Δ*BAT2*) in M1 and M2 were both higher than those of the parental *S. cerevisiae* 38 (*p* ˂ 0.05, *n* = 3). Based on the knowledge of esterification reactions, we deduced that 2-methylbutyl acetate was the derivative product of 2-methyl-1-butanol. It was reported that esters were easily formed by enzymatic condensation of organic acids and alcohols [6,27]. Therefore, 2-methylbutyl acetate should be derived from the esterification of 2-methyl butanol with active acetic acid (or acetyl-CoA). Our results indicated that the loss of the *BAT2* gene provoked an increase in the 2-methylbutyl acetate yield; i.e., *BAT2* may depress the formation of esters when it exists in parental *S. cerevisiae* 38. 

The fourth product positively correlated with Ile addition was glutamine. As Figure 3D shows, the yields of glutamine increased with an increase in Ile in M1 and M2 whether for the mutant *S. cerevisiae* 38 (Δ*BAT2*) or the parental *S. cerevisiae* 38, and were significantly higher than those of the control group. When comparing the yields of glutamine between the mutant *S. cerevisiae* 38 (Δ*BAT2*) and the parental *S. cerevisiae* 38, whether in M1 or M2, the yield of the former was significantly lower than that of the latter, which showed that the loss of the *BAT2* gene significantly reduced the yields of glutamine in M1 and M2. However, the regulatory power of the *BAT2* gene seemed weaker than that of the initial concentrations of Ile. In addition, the difference between the formation of glutamine and the formation of the above products was that glutamine was produced in the control group, which suggested that glutamine should have been a concomitant reaction product rather than a product of Ile, and it may have promoted the decomposition of Ile. According to previous papers [6,28], glutamine was probably the product of α-ketoglutarate receiving foreign amino groups from Ile molecules, which may have promoted Bat2p (encoded by *BAT2*) to function.

#### 3.2.2. Metabolic Regulation of Leu

Analogously, when Leu was used as the sole nitrogen source, the effects of the Ile addition amount in media and the *BAT2* gene deletion on the metabolites were investigated in media M3 (0.5 M Leu+SCD) and M4 (1.0 M Leu+SCD). SCD (without any amino acids) was used as a control. The results of the GC-MS showed that the respective metabolites of Leu were 3-methyl-butyraldehyde, 3-methyl-1-butanol, and 3-methylbutyl acetate, which was consistent with the previous report, except for 3-methylbutyl acetate [29]. These products were isomers of each other with the products of Ile; their respective mass spectrograms are displayed in Appendix A. Their yields are displayed in Figure 4A–C, respectively. In Figure 4A, it can be seen that 3-methyl-butyraldehyde was not detected in the control group, but was detected in M3 and M4 for both the mutant *S. cerevisiae* 38 (Δ*BAT2*) and the parental *S. cerevisiae* 38. The yields of 3-methyl-butyraldehyde reached 10 ± 1.6 μM in M3 and 24 ± 3.9 μM in M4 for the mutant *S. cerevisiae* 38 (Δ*BAT2*), and 63 ± 6.0 μM in M3 and 140 ± 13.0 μM in M4 for the parental *S. cerevisiae* 38, which roughly presented a proportional relationship with the addition amount of Leu. The difference between groups was extremely significant (*p* < 0.01). This showed that the higher the initial concentration of Leu, the higher the yield of 3-methyl-butyraldehyde, and suggested that the initial concentration of Leu upregulated and accelerated its metabolism. When comparing the yields of 3-methyl-butyraldehyde between the mutant *S. cerevisiae* 38 (Δ*BAT2*) and parental *S. cerevisiae* 38, we observed that the deletion of *BAT2* resulted in a significant decrease in the yields of 3-methyl-butyraldehyde in both M3 and M4 (*p* < 0.05). We calculated that the deletion of the *BAT2* gene led to a ~6-fold decrease in the yields of 3-methyl-butyraldehyde.

The second product positively correlated with Leu addition was 3-methyl-1-butanol. As shown in Figure 4B, the yields of 3-methyl-1-butanol in M3 and M4 rose significantly relative to the control group (*p* < 0.01), and reached 0.09 ± 0.006 mM in M3 and 0.44 ± 0.026 mM in M4 for *S. cerevisiae* 38 (Δ*BAT2*), and 1.23 ± 0.062 mM in M3 and 2.17 ± 0.11 mM in M4 for the parental *S. cerevisiae* 38. This showed that the higher the Leu substrate concentration, the greater the 3-methyl-1-butanol yield, based on which we deduced that the initial Leu concentration in vitro upregulated its catabolism level. Between the mutant *S. cerevisiae* 38 (Δ*BAT2*) and the parental *S. cerevisiae* 38, the yields of 3-methyl-1-butanol of the former were significantly lower than the latter’s yields (*p* < 0.01). This indicated that the deletion of *BAT2* significantly led to a 4.93~13.6-fold decline in 3-methyl-1-butanol, which was higher than that of Ile catabolism. We deduced from the above results that 3-methyl-1-butanol was a catabolite of Leu. According to the direction of the Ehrlich pathway [2,5,6], 3-methyl-1-butanol should be converted from 3-methyl-butyraldehyde. This was circumstantiated by the results showing that the yields of 3-methyl-1-butanol also significantly increased with the increase in 3-methyl-butyraldehyde. However, our results showed that the yield of 3-methyl-1-butanol was much higher than that of 3-methyl-butyraldehyde, which may have been due to the aldehyde group of 3-methyl-butyraldehyde being very active, and it was easily reduced to the hydroxyl group and formed 3-methyl-1-butanol. The above results indicated that the initial concentration of Leu and *BAT2* gene jointly upregulated the production of 3-methyl-1-butanol.

The third product positively correlated with Leu addition was 3-methylbutyl acetate. As shown in Figure 4C, 3-methylbutyl acetate was not detected in the control media, but was detected in M3 and M4 for the two strains. This showed that the higher the initial concentration of Leu, the higher the yield of 3-methylbutyl acetate. However, the yields of 3-methylbutyl acetate for the mutant *S. cerevisiae* 38 (Δ*BAT2*) in M3 and M4 were significantly higher than those of the parental *S. cerevisiae* 38, and reached 2.37 ± 0.26 μM in M3 and 4.0 ± 0.017 μM in M4 for the mutant *S. cerevisiae* 38 (Δ*BAT2*), and 0.5 ± 0.003 μM in M3 and 1.0 ± 0.11 μM in M4 for the parental *S. cerevisiae* 38. This indicated that the disruption of the *BAT2* gene promoted the increase in the 3-methylbutyl acetate yield. In other words, the *BAT2* gene may therein depress the formation of acetic acid esters.

The fourth product positively correlated with Leu addition was glutamine. As Figure 4D shows, the yields of glutamine increased with the increase in the concentrations of Leu, whether for the mutant *S. cerevisiae* 38 (Δ*BAT2*) or the parental *S. cerevisiae* 38, in M3 and M4. However, glutamine was also produced in the control group for the two strains, which suggested that glutamine was not the product of Leu; it should have been the accompanying product of the Leu transamination reaction, according to a previous paper [6]. We observed that the deletion of the *BAT2* gene led to a significant decline in the yields of glutamine. This suggested that the deletion of the *BAT2* gene could reduce the intensity of the transamination reaction.

## 4. Discussion

### 4.1. Effects of BAT2 Sequence Difference on Bat2p Activity for Leu and Ile

The sequence of the BAT2 gene of the parental S. cerevisiae 38 was compared with those of a total of 30 S. cerevisiae strains published in the Saccharomyces Genome Database (https://www.yeastgenome.org, 23 January 2021); the results of the sequence alignment analyses are shown in Appendix A. We observed that the coding sequence of the BAT2 gene of the parental S. cerevisiae 38 was the same as those of AWRI1631, AWRI1796, CLIB215, RM11-1a, and YJM789, and the differences with the other 24 strains were presented as single-nucleotide polymorphisms (SNPs) at multiple sites (*n* ≤ 6). We found that there were ~35 same repeats (≥7 bases for every repeat) in the coding sequence (CDS) of the BAT2 gene of the 30 strains; these were regularly distributed in the BAT2 coding sequence of the majority of the yeast strains. The functions of the interspersed repeats remain unclear, but suggest that frequent homologous recombination occurred in the evolution process of the BAT2 gene. The limited base differences in the BAT2 gene sequence only caused two amino acid residues to be altered at positions 41 and 131 of Bat2p (branched-chain-amino-acid transaminase 2), as shown in Appendix A. Relative to the wild-type strain S288C and the other eight commercially transformed strains (BY4741, BY4742, CEN, PK113-7D, Sigma 1278b, W303, EC9-8, and FL100), the amino acid residue at position 41 of the parental S. cerevisiae 38 was valine (V) instead of alanine (A), which overwhelmingly accounted for ~73% of all yeast strains. The amino acid residue at position 131 of the parental S. cerevisiae 38 was tyrosine (Y) instead of Cysteine (C), which overwhelmingly accounted for ~76% of all yeast strains. The total statistical results showed that the homologies of the BAT2 gene and Bat2p were both over 99% among the 30 strains, which suggested that the gene was highly conserved in the S. cerevisiae genome. In order to compare the effects of the BAT2 sequence on the Bat2p hydrophobicity of the parental S. cerevisiae 38, the secondary structural features of Bat2p were modeled along its chain and compared with the Bat2p-288C of wild-type S. cerevisiae 288C using Clone Manager 7.1 software; these are illustrated in Appendix A. We observed that the decision constants of the α-helix and β-sheet around position 41 of the Bat2p-38 chain both were less than those of Bat2p-288C when residue A was replaced by V at position 41, which was consistent with a finding that A is a good helix-forming residue, but V is a poor helix-forming residue [30]. There was no significant difference between the β-turn of the former and that of the latter here. The opposite occurred around position 131 of the Bat2p chain—the decision constants of the α-helix and β-sheet of Bat2p-38 were higher than those of Bat2p-288C. However, the β-turn of the former was less than that of the latter here. The substitution of Y for C may simply have left the protease in a lipid-soluble form [31]. However, the effects of these two site substitutions on the tertiary structure of Bat2p are not known. The hydrophobicity change for Bat2p was estimated using Clone Manager 7.1 software and is shown in Appendix A. We observed that the hydropathic index at around position 41 of Bat2p-38 was closer to the center point than that of Bat2p-288C. This meant that the hydrophilicity of the former was higher than that of the latter here, which was consistent with the results simulated by both the Kyte and Doolittle method (KD) [32,33] or the Hopp and Woods method (HW) [34,35]. Because the two amino acids both had a nonpolar hydrophobic native structure, the hydrophobicity of the substituted valine (value = 1.66) was higher than that of alanine (value = 0.42) [36], which may have resulted in the increase in the hydrophobicity in the region of the former. Here, there was no significant difference for the surface exposure (SE) between the former and the latter. Around the other position 131, the hydropathic index of Bat2p-38 was less than that of Bat2p-288C, because the hydrophobicity of Y (value = 1.31) was slightly less than that of C (value = 1.34) [36]. Meanwhile, the SE of hydrophilic groups around Bat2p-38 was higher than that of Bat2p-288C here. The differences in these secondary structure properties of Bat2p may have affected their tertiary structure and transaminase activity. This may have been due to the different yeast species being forced to adapt to different environments; e.g., the RM11-1a originated from a California vineyard in the USA [37], and the ZTW1 was isolated from a corn mash bioethanol isolate from China [38]. Following the above analyses, we inferred that the hydrophobicity differences at different sites of Bat2p had different activities on Leu and Ile substrates that differed from hydrophobicity. This was verified by the yield difference for 2-methylbutanol with 3-methylbutanol caused by the BAT2 deletion in our results.

### 4.2. Effects of Ile or Leu Addition and BAT2 Gene on High Alcohols and Esters

According to the effects of Ile addition on branched-chain aldehyde, alcohol, and ester flavors, it can be summarized that 2-methyl-butyraldehyde, 2-methyl-1-butanol, and 2-methylbutyl acetate are decomposition products of Ile in the metabolic chain, as shown in Figure 5A, and are all flavors in fermented foods. The chain is a complicated sequential reaction that includes both material flow and energy flow. Presently, some carbohydrate force fields have reached the level that they may even perform better than semiempirical QM methods [39], especially for short-chain organic molecules such as isoamyl alcohol [40]. Based on the MM2 force field, the steric energy of 2-methyl-1-butyraldehyde, 2-methyl-1-butanol, and 2-methyl butyl acetate molecules were calculated to be approximately 5.27 kcal/mol, 5.59 kcal/mol, and 8.50 kcal/mol, respectively. Generally, the smaller the steric energy, the more stable the molecule is. Therefore, we inferred that 2-methyl-1-butyraldehyde may have been the most stable product, followed by 2-methyl-1-butanol, while 2-methyl butyl acetate seemed to be the least stable product among them. Thus, the yield of 2-methyl-1-butyraldehyde should have been the highest, and the yield of 2-methyl butyl acetate should have been the lowest among them. However, the obtained results did not actually reflect this. When comparing Figure 3B with Figure 3A, it can be seen that the yield of 2-methyl-1-butanol was the highest, and was at least an order of magnitude higher than those of 2-methyl-butyraldehyde for the two strains. This indicated that most of 2-methyl-1-butyraldehyde was easily converted to 2-methyl-1-butanol, but in which the participation of other energies may have been required. According to previous reports [2,5,6], NADH/NAD^+^ may be involved in this reaction. Because NADH/NAD^+^ are the most powerful metabolites in metabolic networks in all living organisms, they provide redox carriers for biochemical reactions and act as principal agents in energy transformation inside the cell [41]. Therefore, NADH may impart H^+^ to 2-methyl-1-butyraldehyde and then reduce it to 2-methyl-1-butanol. A strategy to control NADH content or the NADH/NAD^+^ ratio may be effective in controlling the formation of branched higher alcohols. In Figure 5A, it can be observed that the conversion reaction of 2-methyl-1-butanol to 2-methylbutyl acetate should require an higher energy-consuming value than that needed in the conversion reaction of 2-methyl-1-butyraldehyde to 2-methyl-1-butanol, which requires the participation of energy cofactors such as NADPH/NADP^+^, ATP/ADP, and NADH/NAD^+^. So far, there are no reports that these energy cofactors are involved in this reaction. However, it has been reported that acetyl-CoA was involved in the esterification of branched-chain alcohols to form acetate esters with the catalyzation of alcohol acetyltransferase encoded by ATF1 [8,10], but neither of these reports dealt with the energy requirements and sources of this conversion reaction. When comparing Figure 3C with Figure 3A, it can be seen that the yields of 2-methylbutyl acetate of the parental *S. cerevisiae* 38 were lower than those of 2-methyl-butyraldehyde both in M1 and M2. When comparing Figure 3C with Figure 3B, it can be seen that the yields of 2-methylbutyl acetate of the two strains were both significantly lower than those of 2-methyl-1-butanol, whether in M1 or M2. Actually, the yield of 2-methyl butyl acetate was the lowest along the chain, which was consistent with the energy analysis above. The three products were secreted outside the yeast cell and resulted in a relatively stable distribution in the fermentation broth, which indicated that 2-methyl-1-butanol was the most stable metabolite in the metabolic chain, 2-methyl-butyraldehyde ranked second, and 2-methylbutyl acetate was the least stable metabolite. This may have been due to the fact that the aldehyde groups of the branched higher aldehydes were most easily reduced by hydrogenation in the liquid system under the influence of branched methyl groups. It can be summarized that 2-methyl-1-butyraldehyde was regarded as a reservoir of 2-methyl-1-butanol, which resulted in the largest yield of 2-methyl-1-butanol; meanwhile, only a small amount of 2-butylmethanol was acetoxylated to 2-methylbutyl acetate, which should have been a high-energy-consumption process. In other words, there was an energy-storage process for the formation of the acetate ester, in which the excess energy could be stored.

When *BAT2* was deleted, the yield of 2-methyl-butyraldehyde still increased with the addition of Ile, as shown in Figure 3A, but the yield of the mutant strain decreased significantly relative to the parental strain, which suggested that *BAT2* may regulate the formation of part of 2-methyl-butyraldehyde. Why was some 2-methyl-butyraldehyde still produced when *BAT2* was deleted? Through a comparison of the representative TEM images of the two strains in the M1 medium shown in Figure 5B,C, we observed that when the *BAT2* gene was deleted, the distribution density of mitochondria at the cell wall of the mutant was greater than that of the mitochondria near the cell wall of the parental strain. This suggested that 2-methylbutyraldehyde may have been converted from Ile that entered into mitochondria, and the conversion was regulated by Bat1p [9]. The yield of 2-methyl-1-butanol decreased after the deletion of *BAT2*. The reason was that the yield of the 2-methyl-butyraldehyde had been reduced.

When *BAT2* was deleted, the yields of 2-methylbutyl acetate of the mutant *S. cerevisiae* 38 (*ΔBAT2*) increased relative to those of the parental strain in M1 or M2. On the other hand, the presence of *BAT2* depressed the formation of 2-methylbutyl acetate to maintain 2-methyl-1-butanol production at a high level. Therefore, *BAT2* can be inferred to regulate the energy balance of this metabolic chain, since the acetic esterification of 2-methyl-1-butanol is an energy-consuming process [42]. Bat2p may function as an esterase to modulate the balance of the reversible reaction between 2-methylbutyl acetate and 2-methyl-1-butanol, which may be a feedback-regulation mechanism ubiquitous in the formation of the aroma profile of wine, beer, and cider [4,43]. Based on the above analysis and discussion, an improved metabolite chain of Ile was drawn (Figure 5E). It was reported that glutamine was the accompanying product of the Ile transamination reaction prior to the formation of 2-methyl-butyraldehyde [4,6,9]. Our results indicated that glutamine should be a concomitant product of Ile catabolism. Therefore, it corroborated the results of previous studies, as illustrated in Figure 5E.

The effects of Leu addition and *BAT2* on branched-chain aldehyde, alcohol, and ester flavors were very similar to those of Ile. 3-Methyl-butyraldehyde, 3-methyl-1-butanol, and 3-methylbutyl acetate should have been the decomposition products of Leu in the metabolic chain, as shown in Figure 5D. The steric energy values of 3-methyl-1-butyraldehyde, 3-methyl-1-butanol, and 3-methyl butyl acetate molecules were simulated to be approximately 3.81 kcal/mol, 4.36 kcal/mol, and 7.78 kcal/mol, respectively, through the MM2 force field. The metabolite chain also should have been an energy-consumption process or an energy-storage process. For any one strain, the yield of 3-methyl-1-butanol was the largest, that of 3-methyl-1-butyraldehyde was second, and that of 3-methylbutyl acetate was the minimum. 3-Methyl-1-butanol should have been the most stable among the three metabolites. The results suggested that Bat2p encoded by *BAT2* depressed the formation of 3-methylbutyl acetate, and may have regulated both ends of the metabolite chain in a feedback manner. An improved metabolite chain of Leu is shown in Figure 5F.

We concluded that the flux of metabolites in the two metabolic chains accounted for only a small part of the Ile or Leu addition amounts based on the synchronized reverse-phase high-performance liquid chromatography (RP-HPLC) analysis of Ile or Leu depletion (data not shown). Most of the Ile or Leu may have been involved in the synthesis of the yeast’s own proteins through ribosomes. Therefore, there should have been three metabolic flux pathways for the consumption of Ile or Leu, as shown in Figure 5E,F.

The yields of catabolites of Ile or Leu were positively correlated with substrate concentration, so there must exist an unknown signal-aware pathway in the yeast cell [44,45]. Therefore, uncovering this signaling pathway is an important future research subject. In addition, if there is a need to enhance branched-chain higher alcohols and esters in the aroma profile of fermented foods, it is necessary to add enough BCAAs to the fermentation medium. If there is a need to reduce them, three important methods include reducing the added amount of branched-chain amino acids, mutating the *BAT* gene, and eliminating the role of energy cofactors such as NADH/NAD^+^.

A previous report [7] showed that the deletion of both the *BAT2* and *BAT1* genes resulted in an auxotrophic strain of branched-chain amino acids (Ile, Leu, and Val), which indicated that *BAT* genes are important in the life cycle of yeasts. Recently, several new studies showed that *BAT2* played a role in the process of the biosynthesis and transportation of amino acids, and may control the secondary metabolism [3,29,46,47,48,49]. In this work, whether in the 2-methyl-butyraldehyde–2-methyl-1-butanol–2-methylbutyl acetate pathway or the 3-methyl-butyraldehyde–3-methyl-1-butanol–3-methylbutyl acetate pathway, we discovered that the *BAT2* gene upregulated the formation of 2-methyl-butyraldehyde or 3-methyl-butyraldehyde and downregulated the formation of 2-methylbutyl acetate or 3-methylbutyl acetate while maintaining the maximum yield of 2-methyl-1-butanol or 3-methyl-1-butanol, and may have maintained the energy balance in a feedback-regulated manner in the pathway. Therefore, the process should be a reversible reaction [50], as the *BAT2* gene likely regulates the metabolic balance of BCAAs in their decomposition pathways.

## 5. Conclusions

In short, we reached the following conclusions. (1) 2-Methyl-butyraldehyde, 2-methyl-1-butanol, and 2-methylbutyl acetate were the catabolites of Ile; 3-methyl-butyraldehyde, 3-methyl-1-butanol, and 3-methylbutyl acetate were the catabolites of Leu; and their yields were jointly regulated by the initial concentrations of Ile or Leu and *BAT2*, but the regulation of the initial concentrations of Ile or Leu was relatively independent of *BAT2*. (2) The *BAT2* gene is a highly conserved gene in the *Saccharomyces* genome. The *BAT2* gene indirectly upregulated the formation of 2-methyl-butyraldehyde, 2-methyl-1-butanol, 3-methyl-butyraldehyde, and 3-methyl-1-butanol by controlling the transamination reaction. Furthermore, it downregulated the formation of 2-methylbutyl acetate and 3-methylbutyl acetate, in which the *BAT2* gene is speculated to regulate and maintain the decomposition balance of BCAAs. (3) Among the catabolism products, 2-methyl-1-butanol and 3-methyl-1-butanol seemed to be the most stable, and their yields were both the largest in their respective pathways. The yields of 2-methyl-butyraldehyde and 3-methyl-butyraldehyde ranked second, while the yields of 2-methylbutyl acetate and 3-methylbutyl acetate were the smallest among the metabolites.

## Figures and Tables

**Figure 1 genes-13-01178-f001:**
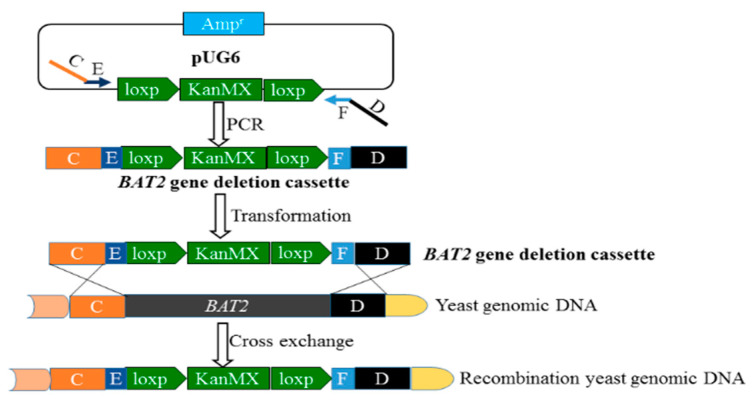
Construction scheme of the *BAT2* deletion cassette using LPSHR.

**Figure 2 genes-13-01178-f002:**
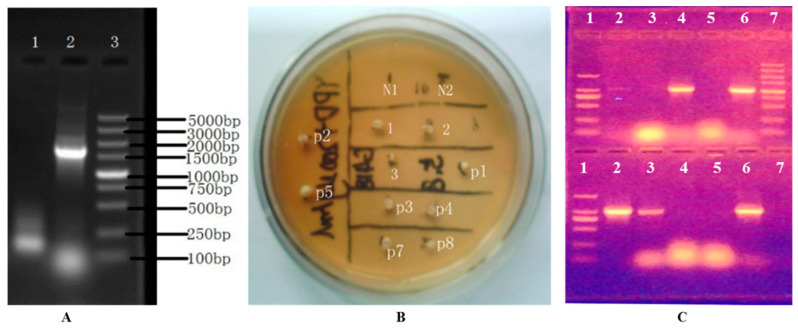
The selection and verification of the deletion cassette of *BAT2* gene and true *BAT2* gene mutants. (**A**) The verification of the constructed *BAT2* gene-deletion cassette. Therein, Lane 1: negative control; Lane 2: *BAT2* gene-deletion cassette; Lane 3: DL 5000 DNA marker. (**B**) The secondary plate selection of the potential transformants using 200 μg/mL of G418. Therein, N1 and N2: negative controls (untransformed colonies); 1, 2, 3, p1, p2, p3, p4, p5, p7, and p8: the 10 potential transformants with G418 resistance. (**C**) The verification of true *BAT2* gene-deletion mutants with PCR and electrophoresis. Therein, upper panel: Lane 1: DL2000 DNA marker (bands from top to bottom are in the order of 2000 bp, 1000 bp, 750 bp, 500 bp, 250 bp, and 100 bp); Lane 2: negative control of PCR; Lanes 3, 4, 5, and 6: the PCR product bands of *BAT2* gene of transformants p1, p3, p4, and p7, respectively; Lane 7: DL5000 DNA marker; lower panel: Lane 1: DL2000 DNA marker; Lane 2: the PCR product of *BAT2* gene of the parental *S. cerevisiae* 38; Lanes 3, 4, 5, and 6: the PCR product bands of *BAT2* gene of transformants 1, 2, 3, and p8, respectively.

**Figure 3 genes-13-01178-f003:**
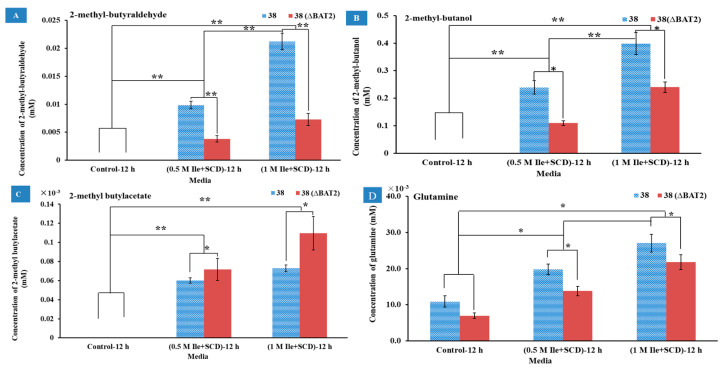
The effects of Ile addition amount in media and *BAT2* gene deletion on the metabolites. (**A**) The effects of Ile addition amount in media and *BAT2* gene deletion on the yields of 2-methyl-butyraldehyde; (**B**) the effects of Ile addition amount in media and *BAT2* gene deletion on the yields of 2-methyl-1-butanol; (**C**) the effects of Ile addition amount in media and *BAT2* gene deletion on the yields of 2-methylbutyl acetate; (**D**) the effects of Ile addition amount in media and *BAT2* gene deletion on the yields of glutamine. ** Extremely significant (*p* ˂ 0.01, *n* = 3); * significant (*p* ˂ 0.05, *n* = 3).

**Figure 4 genes-13-01178-f004:**
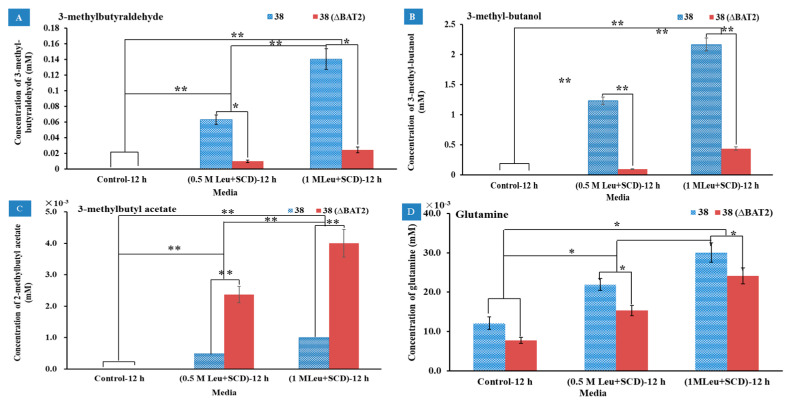
The effects of Leu addition amount in media and *BAT2* gene deletion on the metabolites. (**A**) The effects of Leu addition amount in media and *BAT2* gene deletion on the yields of 3-methyl-butyraldehyde; (**B**) the effects of Leu addition amount in media and *BAT2* gene deletion on the yields of 3-methyl-1-butanol; (**C**) the effects of Leu addition amount in media and *BAT2* gene deletion on the yields of 3-methylbutyl acetate; (**D**) the effects of Leu addition amount in media and *BAT2* gene deletion on the yields of glutamine. ** Extremely significant (*p* ˂ 0.01, *n* = 3); * significant (*p* ˂ 0.05, *n* = 3).

**Figure 5 genes-13-01178-f005:**
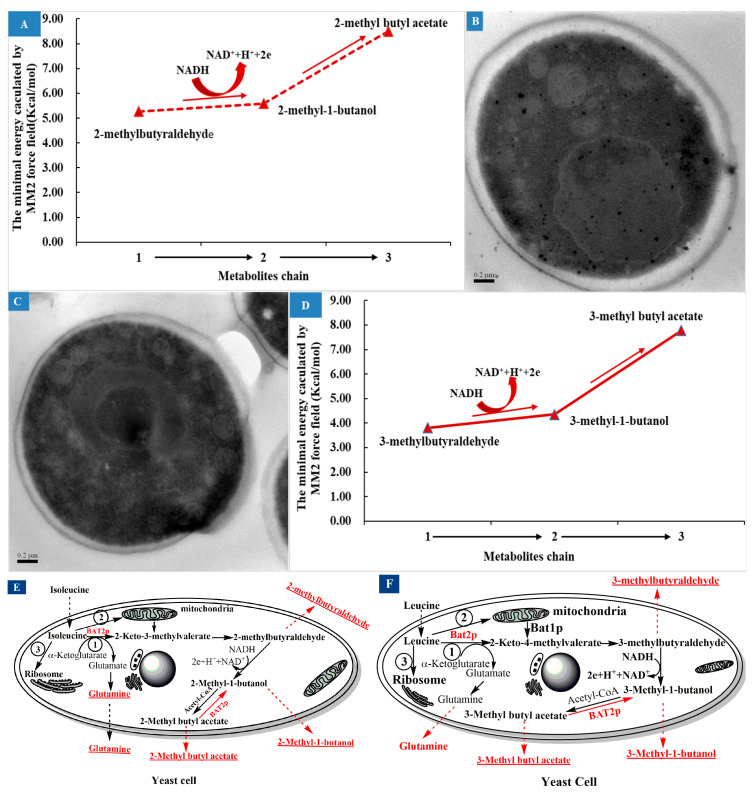
The metabolite chain analysis of Ile and the potential effects of *BAT2* gene on the chain. (**A**) The simulation analysis of the substance and energy metabolism chain of Ile. (**B**) A representative transmission electron microscope photo of parental *S. cerevisiae* 38 in M1 (0.5 M Ile+SCD). (**C**) A representative transmission electron microscope photo of *BAT2* mutant 38 in M1 (0.5 M Ile+SCD). (**D**) The simulation analysis of the substance and energy metabolism chain of Leu. (**E**) The improved catabolism pathway of Ile and Bat2p regulation. (**F**) The improved catabolism pathway of Leu and Bat2p regulation.

**Table 1 genes-13-01178-t001:** Primers used in this study.

No.	Primers	Sequence (5′→3′)
1	Fw-1	ATTTGCGGCCGCTATCTAATCTGTAGATCCGACT
2	Rv-2	ATTTGCGGCCGCCTTCTAAGGTATGTATGGGC
3	Bat2-deletion-FW	ACCCGTCTCCCCTCAAGATACCAGCATTGCTCCCTCCAACTACTCCAGCTGAAGCTTCGTACGC
4	Bat2-deletion-RV	CTGATAGGCCAGCACTAGATGACAAGAAAAAAAACGAAAGGATAAGCATAGGCCACTAGTGGATCTG
5	fw-bat2-2	TCTAAGCCAAAACCGAAC
6	rv-bat2-2	CTTGACCAATTGCCATGC

## Data Availability

Not applicable.

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
