# Peer review of "The Effects of Catabolism Relationships of Leucine and Isoleucine with BAT2 Gene of Saccharomyces cerevisiae on High Alcohols and Esters"

_genes, 2022, doi:10.3390/genes13071178_

Round 1

Reviewer 1 Report

The English language has grammatical errors and should be reviewed and corrected. Some results are not included in the Result section and are presented just in Discussion section.

31-34: correct easters into esters and rewrite these two sentences to be more clear and understandable.

41: remove one the before degradation, it is doubled

53: remove one modern as it is dobled – in modern modern fermentation industry

63: remove one is in: esters is is lacking

64-67: rewrite the sentence as it is unclear

71: correct S. Cerevisiae into S. cerevisiae

77: add name of the species for the strain 38

127: add space before 1 min

175: add space between pH and 5.5

233: correct Ture into True

254-255: rewrite the sentence to make it more clear

264-281: rewrite this part and clearly explain the consequences of BAT2 deletion on 2-methyl-butyraldehyde production. Probaly in line 279 is refferd to 2-methyl-butyraldehyde and not 3-methyl-butyraldehyde

357: add space between Figure 4B

399: results missing under Result section

451: results missing under Result section

585: correct saccharomyces into Saccharomyces

Author Response

Response to Reviewer 1 Comments

Point 1: Extensive editing of English language and style required

Response 1: Thanks for your valuable comment. We have extensively revised and polished the whole manuscript, now it has been improved substantially for better clarity and readability and made it easily understand as soon as possible. Thank the reviewer’s constructive comments.

Point 2: Are the methods adequately described? It can be improved.

Response 2: The methods have been described adequately and improved than before.

Point 3: Are the results clearly presented? Must be improved.

Response 3: The presented results have been revised carefully and now improved better than before.

Point 4: Are the conclusions supported by the results? It can be improved.

Response 4: We have revised the conclusions section and improved it much better than before.

Point 5: The English language has grammatical errors and should be reviewed and corrected. Some results are not included in the Result section and are presented just in Discussion section.

Response 5: We have revised word for word and corrected the English grammar errors of the whole manuscript. Some results about molecular energy and electron microscopy were presented in Discussion section in order to be convenient to discuss and draw the improved catabolism pathway of Ile and Leu. After carefull consideration, we want to maintain the style of writing here because it is easy for readers to understand.

Point 6: 31-34: correct easters into esters and rewrite these two sentences to be more clear and understandable.

Response 6: Thanks the reviewer’s careful comments. We have corrected easters to esters and rewritten these two sentences. Please refer to Line 31-34 and list as follows: Branched-chain higher alcohols such as isoamyl alcohol, isobutanol, and isoamyl acetate are the key flavor components and eaters in many fermented foods. Appropriate content of higher alcohols and esters makes the fermented foods mellow, soft, full and harmonious.

Point 7: 41: remove one the before degradation, it is doubled

Response 7: We have removed one the before degradation accordingly. Please refer to Line 42.

Point 8: 53: remove one modern as it is dobled – in modern modern fermentation industry

Response 8: Thanks for your comment. We have removed one modern according the reviewer’s suggestion. Please refer to Line 53.

Point 9: 63: remove one is in: esters is is lacking

Response 9: It has been revised here. Thanks the reviewer’s careful comments. Please refer to Line 64.

Point 10: 64-67: rewrite the sentence as it is unclear

Response 10: Thanks the Reviewer’s constructive comments. We have rewritten the sentence as. Please refer to Line 64-67 and list as follows: In this work, the effects of Ile or Leu addition at high levels and BAT2 deletion on high alcohols and esters derived from Ile or Leu were investigated by respectively using Ile or Leu as the sole nitrogen source of both a BAT2 mutant S. cerevisiae 38 and its Parental S. cerevisiae 38, in which the material flow and energy flow were reasoned.

Point 11: 71: correct S. Cerevisiae into S. cerevisiae

Response 11: We have corrected S. Cerevisiae to S. cerevisiae in the whole manuscript including references list.

Point 12: 77: add name of the species for the strain 38

Response 12: It has been revised accordingly in the whole manuscript.

Point 13: 127: add space before 1 min

Response 13: We have added space before 1 min. Please refer to Line 127.

Point 14: 175: add space between pH and 5.5

Response 14: We have added space between pH and 5.5. Please refer to Line 174. Thanks the Reviewer’s careful comments.

Point 15: 233: correct Ture into True

Response 15: We have corrected Ture into True. Please refer to Line 232. Thanks the Reviewer’s careful comments.

Point 16: 254-255: rewrite the sentence to make it more clear

Response 16: We have rewritten the sentence according to the reviewer’s suggestion. Now the sentence is more concise and clear to read than before. Please refer to Line 252-256 and list as follows: The catabolic analyses of Ile were qualitatively and quantitatively analyzed by GC-MS through comparing metabolites produced by the BAT2 mutant S. cerevisiae 38 (ΔBAT2) and Parental S. cerevisiae 38 at 12 h in media M1 (0.5 M Ile+SCD without other amino acids), M2 (1.0 M Ile+SCD without other amino acids) and control group (SCD without any amino acids).

Point 17: 264-281: rewrite this part and clearly explain the consequences of BAT2 deletion on 2-methyl-butyraldehyde production. Probaly in line 279 is refferd to 2-methyl-butyraldehyde and not 3-methyl-butyraldehyde

Response 17: Thanks for your careful comments. We have rewritten this part and more clearly explained the consequences of BAT2 deletion on 2-methyl-butyraldehyde production than before. Please refer to Line 258-285 and list as follows: As Figure 3A shown, 2-methyl-butyraldehyde was not detected in the control group of the two strains, but which was detected in M1 and M2 groups. The mass spectrogram of 2-methylbutyraldehyde was displayed as Figure S1 (1) in Supplementary material. It can be observed that the 2-methyl-butyraldehyde yields of S. cerevisiae 38 (ΔBAT2) and Parental S. cerevisiae 38 significantly increase with the initial concentration of Ile in both M1 and M2 compared to the control group (P<0.01), which roughly present a proportional relationship between the yield of 2-methyl-butyraldehyde and the addition amount of Ile. On the other hand, the 2-methyl-butyraldehyde yields of S. cerevisiae 38 (ΔBAT2) are significantly less than those of Parental S. cerevisiae 38 in M1 and M2 (P˂0.01, n=3), which are 3.81 ± 0.57 μM in M1 and 7.3 ± 1.01 μM in M2 for S. cerevisiae 38 (ΔBAT2), 9.87 ± 0.69 μM in M1 and 21.2 ± 1.49 μM in M2 for Parental S. cerevisiae 38, respectively. It indicates that the deletion of BAT2 leads to the decrease of the yields of 2-methyl-butyraldehyde, the existence of BAT2 gene upregulates the conversion rate of Ile to 2-methyl-butyraldehyde. It can be calculated that the deletion of BAT2 gene leads to 2.59~2.9 folds decrease of the yields of 2-methyl-butyraldehyde. The finding is consistent with the previous reports [2, 4]. The regulation of initial Ile concentration appears to be independent of regulation of the BAT2 gene. Because the tendency is always that the higher initial concentration of Ile in vitro of yeast cells, the higher yields of 2-methyl-butyraldehyde for the two strains. It seems to have nothing to do with the existence of the BAT2 gene. Which suggests the initial concentration of Ile may play a critical role to give a pressure signal to cell in vitro. This is consistent with the finding that the positive effect of BCAAs addition on major volatile compounds in wine were largely associated with the increment of amino acid transportation (up-regulation of GAP) and yeast population [8]. These results indicated that the initial concentration of Ile and BAT2 gene jointly upregulate the production of 2-methyl-butyraldehyde in Parental S. cerevisiae 38. In short, it can be inferred from the above results that 2-methyl-butyraldehyde is one of decomposition products of Ile, which is consistent with the previous findings [6, 9, 25, 26].

In addition, line 279 has been modified to 2-methyl-butyraldehyde. Please refer to Line 282.

Point 18: 357: add space between Figure 4B

Response 18: We have added space between Figure 4B. Thanks the Reviewer’s careful comments.

Point 19: 399: results missing under Result section

Response 19: Thanks for your comments. We have revised the topic of the section and made it corresponding Result section. While we have revised the section and made it better than before. Please refer to 4.1 Effects of BAT2 sequence difference on Bat2p activity to Leu and Ile  in Line 414-467.

Point 20: 451: results missing under Result section

Response 20: We have revised the topic of the section and made it corresponding Result section. While we have revised the section and made it better than before. Please refer to 4.2 Effects of Ile or Leu addition and BAT2 gene on high alcohols and esters in Line 469-595.

Point 21: 585: correct saccharomyces into Saccharomyces

Response 21: We have corrected saccharomyces into Saccharomyces in the whole manuscript. Thanks for your careful comments.

Reviewer 2 Report

This paper is well written, moderate English changed are required. Results are in accordance with the goals.

Minor comments:

Introduction: Its well written and clear.

Line 71. S. cerevisiae (correct the capital letter in the specific name)

M&M

Line 77: A cider yeast strain 38 used was... please specify the name of the species (S. cerevisiae 38). Same through the text (line 94, 118, ecc).

Line 96 centrifugation units g and in line 180 in rpm, please uniformize units (g or rpm).

Figure 3: Escales of the concentrations should be uniformized or to be transformed into a Heatmap. Same to Figure 4.

Figure 5 is not necessary. It can be presented as supplementary material

Author Response

Response to Reviewer 2 Comments

Point 1: Moderate English changes required.

Response 1: We have revised English language of the whole manuscript. Moreover, we invited a native English speaker to polish the whole manuscript again. Now we believe that the revision has met the publication standard of Genes. Thanks the reviewer’s constructive comments.

Point 2: This paper is well written, moderate English changed are required. Results are in accordance with the goals.

Response 2: We have revised English language of the manuscript and made it better than before.

Point 3: Introduction: Its well written and clear.

Response 3: Thank the reviewer’s active comments.

Point 4: Line 71. S. cerevisiae (correct the capital letter in the specific name)

Response 4: We have corrected the capital letter of S. cerevisiae in the whole manuscript including references list.

Point 5: Line 77: A cider yeast strain 38 used was... please specify the name of the species (S. cerevisiae 38). Same through the text (line 94, 118, ecc).

Response 5: It has been revised accordingly in the whole manuscript. Thank the reviewer’s careful comments.

Point 6: Line 96 centrifugation units g and in line 180 in rpm, please uniformize units (g or rpm).

Response 6: Thanks for your valuable suggestion. We have unified the units into rpm. Please refer to Line 95,140,146 and149.

Point 7: Figure 3: Escales of the concentrations should be uniformized or to be transformed into a Heatmap. Same to Figure 4.

Response 7: We have unified the concentration unit to mM. Please refer to new Figure 3 and Figure 4. Thank you for your rigorous comments.

Point 8: Figure 5 is not necessary. It can be presented as supplementary material.

Response 8: Thanks for your careful scrutiny. We have presented Figure 5 as supplementary material. Please refer to Figure S2 in supplementary material.

Round 2

Reviewer 1 Report

I agree with the author's responses and suggest accepting the article in its present form.